# Electron glass effects in amorphous NbSi films

Julien Delahaye[1], Thierry Grenet[1], Claire A. Marrache-Kikuchi[2],
Vincent Humbert[2], Laurent Bergé[2] and Louis Dumoulin[2]

**1** Univ. Grenoble Alpes, CNRS, Institut Néel, 38000 Grenoble, France
**2** CSNSM, Université Paris-Sud, Orsay, F-91405, France

## Abstract

We report on non equilibrium field effect in insulating amorphous NbSi thin films having different Nb contents and thicknesses. The hallmark of an electron glass, namely the logarithmic growth of a memory dip in conductance versus gate voltage curves, is observed in all the films after a cooling from room temperature to 4.2 K. A very rich phenomenology is demonstrated. While the memory dip width is found to strongly vary with the film parameters, as was also observed in amorphous indium oxide films, screening lengths and temperature dependence of the dynamics are closer to what is observed in granular Al films. Our results demonstrate that the differentiation between continuous and discontinuous systems is not relevant to understand the discrepancies reported between various systems in the electron glass features. We suggest instead that they are not of fundamental nature and stem from differences in the protocols used and in the electrical inhomogeneity length scales within each material.


# 1   Introduction

In the past two decades, intriguing out-of-equilibrium phenomena have been reported in the electrical conductance $G$ of several disordered insulating systems [1]. After they are cooled from room temperature to liquid He, $G$ decreases logarithmically with the time elapsed since the cooling [2, 3]. In most cases no saturation to some equilibrium value can be observed, even after weeks of measurements. A fruitful way to investigate this effect is to make MOSFET structures whose (weakly) conducting channels are thins films of the materials under study. After such a device has been allowed to relax for some time under a given gate voltage ($V_g$) at constant temperature, a change of $V_g$ triggers an immediate increase of the conductance, followed by a new slow downward relaxation. Fast $V_g$ scans show that the relaxation history remains printed in the $G(V_g)$ curves as memory dips (MDs) centred on $V_g$ values at which the device was allowed to relax for some time [2, 4]. MDs are slowly erased with time when $V_g$ is maintained out of these values. In granular Al films it was demonstrated that the conductance relaxation triggered by a $V_g$ change to a pristine value is not exactly logarithmic with time and depends on the total time elapsed since the cooling [5]. This age dependence of the dynamics or ageing effect underlines the strong similarity between these out-of equilibrium phenomena and what is seen in structural and spin glasses.

The origin of the glassy features of disordered insulators has been the subject of many experimental and theoretical efforts. It was soon suggested that they may reflect the existence of an electron glass, a glassy state of the carriers first predicted in the 80's and induced by the coexistence of disorder and ill-screened electron-electron interactions [6–8]. Many numerical works have studied the out-of-equilibrium physics of the Efros-Shlovskii Coulomb glass and related models of disordered insulators [1]. Mean field approaches have suggested a glass phase transition concomitant with the formation of the Coulomb gap which needs the absence of screening present in the glass phase in order to fully develop [9,10]. The occurrence of such a transition was recently found in numerical simulations of the 3D Coulomb glass model [11] but with a transition temperature much smaller than the mean field prediction.

The electron glass scenario was recently strengthened in amorphous indium oxide (a-InOx) films by the demonstration that the duration of the logarithmic relaxation depends on the charge carrier density $n$ (as estimated from room temperature Hall effect) and electrical resistance of the samples. The higher time cutoff of the logarithmic relaxation could be reduced down to measurable times in the low doping and low resistance (sheet resistance $R_s$ close to quantum resistance $R_Q$) regime of insulating InOx films [12]. This offers a clear demonstration of a correlation between the electronic properties and the relaxation time distribution.

But other experimental issues remain controversial or are waiting for a satisfactory explanation. For example, the connection between the MD and Coulomb correlations still need to be clarified. In InOx films, the MD width was found to be insensitive to the sample resistance but increases systematically with $n$ [13]. It was proposed that this $n$ dependence is universal [14] and includes Be [15], $Tl_2O_{3-x}$ [16] and Ge-Te alloys films [17–19], although the carrier concentration cannot be conveniently changed in all of them. Using a percolation approach on a classical disordered model, Lebanon and Muller [20] were able to reproduce the $n$ dependence of the MD width but not its temperature dependence experimentally observed at low temperature in granular Al films [2, 21].

The controversial results reported on the temperature dependence of the glassy dynamics also raise serious questions. In discontinuous metal films (gold, etc.), a marked slowdown upon cooling was highlighted, giving rise to the formation of large frozen MDs [22–24]. More recently, we demonstrated that at low temperature the MD dynamics of granular Al films is thermally activated [25]. In contrast, in the case of InOx, it was argued that the dynamics is not activated and may even accelerate upon cooling in the less doped samples, which was

explained by the quantum nature of the glass [26]. However, the protocols used to quantify the dynamics are more indirect [13, 26] and we have questioned their relevance [27].

Another debated issue has recently emerged about the length over which the glassy features are perturbed upon the application of a gate voltage change, hereinafter called *the penetration length*. In granular Al films, only the 10 nm-thick layer of the film closest to the gate insulator is electrically disturbed by a $V_g$ change [28], while the disturbance extends to more than 70 nm in a-InOx films [29], indicating markedly different length scales in the two systems. The large penetration length observed in InOx films was taken as an experimental evidence for electronic avalanches found after a charge injection in numerical simulations of Coulomb glass models [30,31]. However, they seem to contradict the very short screening lengths sometimes invoked in this system [13, 29].

In light of these contradictory results, one may wonder whether the glassy physics could have a different origin in discontinuous/granular systems and continuous disordered systems like a-InOx, or if the observed differences are specific to the protocols used or to peculiar characteristics of InOx. To answer this question, it is of prime importance to study other amorphous systems which electrical properties can be tuned within a significant range by changing the thickness or the chemical composition of the films. This is what is done in the present study. In a preliminary work on a-NbSi films, we found a strong slow down of the dynamics and the freezing of prominent broad MDs upon cooling from room temperature [32], evidencing activated dynamics in a continuous amorphous system. Here we report on a comprehensive study of electron glassiness in a series of NbSi films of different Nb contents and thicknesses which confirms the general character of the activated dynamics and shed some light on the glassy phenomenology in this system.

The paper is organized as follows. The measurement set-up and film parameters are presented first. Then, we discuss the penetration lengths and how the MD width depends on the film parameters. Last, time and temperature dependencies of the glassy dynamics are explored in detail. The NbSi results are discussed in light of what has been found in other disordered insulating systems.

## 2 Experimental method

The studied a-NbSi films were synthesized by the co-deposition of Nb and Si on substrates at room temperature and under ultra-high vacuum (typically a few $10^{-8}$ mbar) as described elsewhere [33]. Structural investigations have shown that such films are amorphous and continuous at least down to a thickness of $\simeq 2$ nm [33]. In order to perform field effect measurements, the films were deposited onto highly doped Si wafers (the gate) coated with a 100 nm-thick layer of thermally grown $SiO_2$ (the gate insulator). Shadow masks were used to define a film of typical size 0.6 mm × 1.6 mm. The films were contacted through electrodes made of 3 nm of Ti for adherence and 5 nm of Pd. Most films were protected from oxidation, in-situ, by a 12.5 nm-thick SiO overlayer. The SiO overlayer was found to play no significant role in the glassy behavior described below [32].

The electrical conductance was measured in a two-point contact configuration. An AC or a DC bias voltage is applied and the resulting current is detected using a low noise current amplifier. The bias voltage was kept sufficiently small to remain in the ohmic regime (in AC, the bias voltage should also be much smaller than the $V_g$ width of the MD). $V_g$ was allowed to vary between -30 V and 30 V, far enough from the practical breakdown limit of our $SiO_2$ gate insulating barrier (around 50 V). No significant leaking current can be measured within this $V_g$ range.

Six samples with two different thicknesses $T_h$ (2.5 nm and 12.5 nm) and three differ-

Table 1: Names and parameters of the a-NbSi films used in this study.

| Name | Thickness | % Nb | $R_{s300K}$ | $R_{s4K}$ | $RRR$ |
|---|---|---|---|---|---|
| A1 | 2.5 nm | 13 | 16.8 kΩ | 97 MΩ | 5800 |
| A2 | 2.5 nm | 14 | 13.0 kΩ | 7.0 MΩ | 540 |
| A3 | 2.5 nm | 16 | 10.1 kΩ | 330 kΩ | 33 |
| B1 | 12.5 nm | 7 | 10.7 kΩ | 53 MΩ | 5000 |
| B2 | 12.5 nm | 8 | 7.2 kΩ | 970 kΩ | 135 |
| B3 | 12.5 nm | 10 | 4.3 kΩ | 28 kΩ | 6.5 |

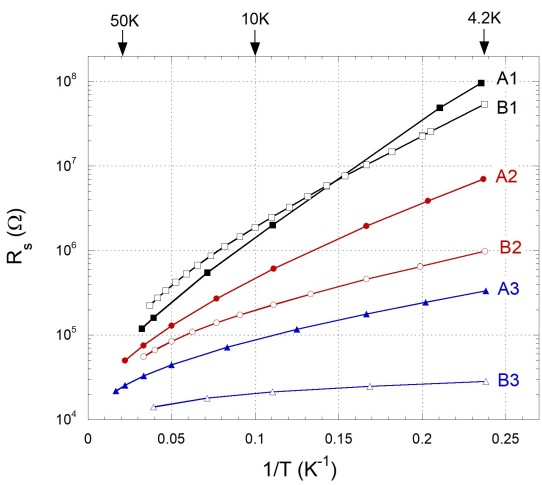

Figure 1: Log of $R_s$ versus $1/T$ between $\simeq$ 50 K and 4.2 K for the six a-NbSi films used in this study. The lines are guides for the eye. Full symbols: 2.5 nm thick films; empty symbols: 12.5 nm thick films.

ent Nb contents for each thickness were measured. Their names used hereafter and their general electrical parameters (resistance per square $R_s$ at 300 K and 4.2 K, resistance ratio $RRR = R_{4K}/R_{300K}$) are given in Table 1.

The electrical properties of a-NbSi samples synthesized by e-gun deposition have been studied by different authors [34–36]. When the films thickness exceeds $\simeq$ 100 nm (3D samples), the films are metallic or superconducting when the Nb content is above $\simeq$ 9% and insulating when it is below. When the thickness is smaller than 100 nm, the insulating state is observed up to larger Nb concentration: up to $\simeq$ 14% for a thickness of 12.5 nm and up to $\simeq$ 20% for a thickness of 2.5 nm. All the samples measured in this study thus lie on the insulating side of the metal- or superconductor-to-insulator transition. The insulating character of the films is confirmed by their resistance versus temperature dependence. In Figure 1, we show Arrhenius plots of the low T parts of the samples' resistances. In the range 4 K - 50 K, they are well described by an exponential-like divergence $R \propto \exp(T_0/T)^\alpha$ with an exponent $\alpha$ which varies between 0.5 and 0.8 depending on the samples, a behaviour commonly observed in disordered insulating materials [25, 34, 37].

# 3 Conductance relaxations after a cool down to 4.2 K: overall behaviour

Let us first present how the samples conductance evolves after they are cooled down to liquid He under a fixed $V_g$. The films are first cooled from room temperature to 4.2 K, which takes about 15 min with our experimental set-up. During the cooling, $V_g$ is kept constant and equal to $V_{geq}$. Once at 4.2 K, $G$ is measured as a function of $V_g$ as sweeps are performed around $V_{geq}$. Between each sweep, $V_g$ is maintained at $V_{geq}$ for a waiting time at least ten times as long as the sweep duration. The conductance variations induced by the small temperature drifts of the He bath are corrected [38] and we end up with the set of $G(V_g, t)$ curves plotted in Figure 2.

Beyond the growth of prominent MDs centred on $V_{geq}$, what is remarkable is the wide variety of $G(V_g)$ shapes and evolutions observed for the different films. First, an overall downward drift of the curves called hereinafter *background relaxation* is observed to various degrees, from almost absent in the 2.5 nm-thick films to very pronounced in 12.5 nm-thick films. Second, the MD width becomes significantly larger when the $R_s$ value decreases. We now discuss these two features and their possible significance.

# 4 Background relaxations and screening length values

The existence of a background relaxation can be best visualized by plotting $G(t)$ curves at different $V_g$ from the data of Figure 2, as shown in Figure 3. In 2.5 nm thick films, $G$ relaxations are significant at $V_{geq}$ but much smaller far from it ($G$ is even constant at 30 V in sample A1, left panel of Figure 3). In 12.5 nm thick films however, the relaxations remain large even 30 V away from $V_{geq}$ clearly demonstrating the background relaxation effect.

Similar features were also observed in granular Al films [28] and are believed to show up when the static screening length $L_{sc}$ becomes smaller than the film thickness $T_h$. The evolution of the conductance observed at a given $V_g$ can then be seen as the sum of two separate contributions: one coming from the part of the film within $L_{sc}$ from the gate insulator which is influenced by the applied $V_g$ (the $V_g$-sensitive layer), and another one coming from the rest of the film which is insensitive to $V_g$ changes (the $V_g$-insensitive layer) and which pursues it relaxation induced by the initial cooling. Note that according to this interpretation, the penetration length introduced earlier is nothing else than the screening length of the system. As long as the conductance is measured at $V_{geq}$, both layers contribute to the observed relaxation. But for the conductance observed at a $V_g$ value far enough from $V_{geq}$, the $V_g$-sensitive layer equilibrated at $V_{geq}$ has its excited and time-independent conductance (see the sample A1 at 30 V in Figures 2 and 3), so that the only contribution to the relaxation that remains is the one of the $V_g$-insensitive layer. The relaxation of the $V_g$-sensitive layer results in the growth of the MD at $V_{geq}$. Assuming that the relaxation modes are homogeneously distributed throughout the thickness of the films (see Figure 4), the $G$ vs. $\ln(t)$ relaxation slopes should be proportional to the thickness of the relaxing layer, i.e. $T_h$ when measured at $V_{geq}$ and $(T_h - L_{sc})$ far enough from $V_{geq}$ when the slope is (almost) $V_g$-independent [28]. By comparing the $G$ vs. $\ln(t)$ relaxation slopes at $V_{geq} = 0V$ and at $30V$ in our 12.5 nm thick films, we get $L_{sc}$ estimates of 1.3 - 1.9 nm for sample B3, 4 nm for B2 and 7 nm for B1 (the interatomic distance $d_{Nb-Si}$ is around 0.26 nm [36]). In 2.5 nm thick films, the relaxation slope at 30 V is at least 5 times smaller than at 0 V, which means that $L_{sc}$ is of the order or larger than 2.5 nm. In granular Al films, a screening length of about 10 nm was extracted from 20 and 100 nm thick films background relaxations, which was found to be roughly constant with $R_s$ in the samples investigated, probably due to the granular nature of the system [28].



Figure 2: Conductance versus $V_g$ curves measured at 4.2 K and at times $t$ after a cooling from room temperature. $V_{geq}$ was fixed to 0 V during and after the cooling for all the samples, and the total sweep time was at least 10 times smaller than the waiting time between two sweeps. Left column: 2.5 nm thick films; right column: 12.5 nm thick films. Refer to Table 1 for the samples parameters.

In InOx films, no background relaxations were reported even in films as thick as 75 nm [29], which in our picture would imply significantly larger screening lengths. Several other experimental results shed some light on this specificity of InOx films as compared to other systems. Transmission electron microscope investigations in NbSi and InOx films have shown that, although these two systems are amorphous from the structural point of view, their chemical disorder length scales are markedly different. NbSi thin films are highly homogeneous,

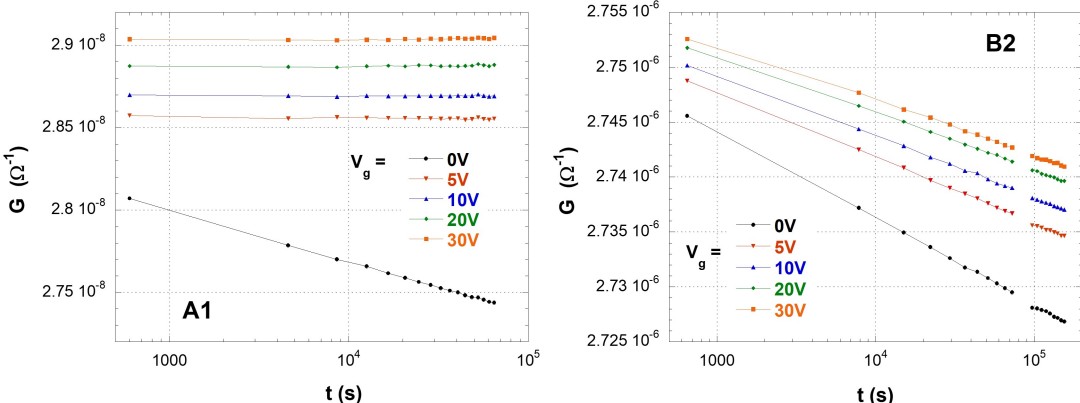

Figure 3: Two examples of $G(t)$ sets of curves measured at different $V_g$ values at 4.2 K after a cooling from room temperature under $V_{geq} = 0$ V (same data as Figure 2). Left side panel: sample A1; Right side panel: B2.

with compositional fluctuations of less than 0.1% down to 1 nm [36]. By comparison, the variations of the O/In ratio in InOx films can be as large as 15-40% for a sampling area of 5 nm × 5 nm and these compositional fluctuations persist over scales extending up to 30 - 80 nm [39], which should result in large conductance fluctuations over length scales of the same order. Furthermore, a qualitatively similar difference between granular Al and InOx films can be inferred from resistor network percolation radius estimates, which are as large as 300 nm in InOx films [40] compared to 30 nm in granular Al films with similar $R_s$ values [41]. The bias voltage extent of the ohmic regime is found to be systematically smaller in InOx films as compared to the granular Al case [2,29,40] and points in the same direction. More recently, the smallness of the volume occupied by the current carrying network has been outlined in InOx films [42], most of the remaining volume being attributed to (much) more insulating zones where the very slow electron dynamics is believed to take place. In a MOSFET device, these highly insulating zones will allow the penetration of the unscreened electrical field over large distances. All these findings suggest that a material specific long range electrical inhomogeneity may explain why the penetration length is larger in InOx than in granular Al and NbSi films.

In contradiction with the preceding, it was suggested that in InOx films, the screening length is as small as $\simeq 1$nm [29]. This estimate relies on a metallic-like screening model and a charge carrier density given by the Hall effect measurement at room temperature. But according to theoretical studies, screening in a system of disordered localized electrons is very different from what it is in a metal, especially at low $T$. Mean field approaches predict that screening is suppressed at low $T$ in the glassy phase, which preserves the long-range part of Coulomb interactions and allow the opening of the Efros-Shklovskii Coulomb gap [10,43]. When thermal effect and Coulomb interactions compete, the density-of-states at the Fermi level remains finite and the naive application of Thomas-Fermi theory gives a screening length that goes like $1/T$ [43]. Numerical simulations at $T = 0$ found that a charge injection in an electron glass triggers an electronic avalanche which size scales with the size of the system [30,31]. This divergence of the screening length at low $T$ was actually experimentally inferred from the $T$ study of the background relaxation in granular Al films [38]. It thus seems dubious to us that very short screening lengths exist at low temperature in insulating InOx films. These controversial results emphasize that the physical origin of the penetration length, and especially of the large values observed in InOx films, remains an open and debated issue. Screening length determination by other means than gate voltage induced conductance

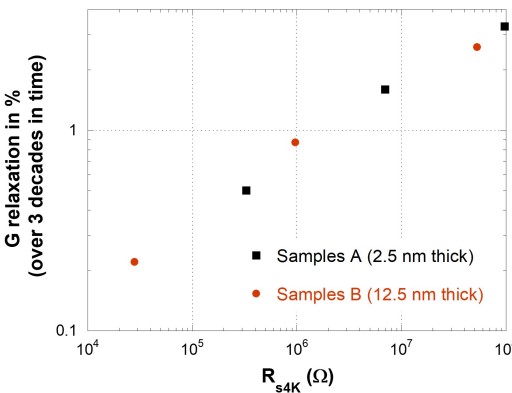

Figure 4: Relative amplitude of the $\ln(t)$ conductance relaxations measured at $V_{geq}$ after a quench to 4.2 K over three decades in time as a function of $R_{s4K}$.

relaxations would be very helpful in order to confirm our interpretation.

We end this section with a note on relaxation amplitudes. It is seen in Figure 4 that the amplitudes in % of the $G$ relaxations at $V_{geq}$ after a quench to 4.2 K are essentially determined by the $R_s$ value and not by the Nb content or the thickness of the films. The larger $R_s$, the larger the amplitude. From the above, the relaxation at $V_{geq}$ is the sum of the ongoing MD growth and the background relaxation and it thus reflects the amplitude relaxation of whole volume of the sample. The fact that all the points fall on a smooth curve although the samples have different $T_h/L_{sc}$ values, shows that the relaxing modes are distributed in the whole film thickness, ruling out mechanisms like e.g. charge relaxation in the substrate or at the interface. Note that this $R_s$ trend is common to all the disordered systems in which MDs and conductance relaxations have been observed [1], even if the charge carrier density was also found to play a role in the MD amplitude of InOx films [12].

## 5   Width of the memory dip at 4.2 K

Let us now focus on the MD centred on $V_{geq}$ that is visible for all the films in Figure 2. It actually results from the growth of a narrow 4.2 K contribution (hereafter called the 4.2 K MD) on top of a broad and time independent contribution inherited from the cooling history of the sample [32]. If we plot the differences between the $G(V_g)$ curve measured at the longest time and the ones measured at different times after cooling, they can be superposed for each sample on a single curve simply by using an ad hoc vertical scaling (see Figure 5). This shows that the evolution of the total MD at $V_{geq}$ does not result from a slow "thermalization" of the broader MD formed during the cooling of the samples but from the growth of the 4.2 K MD on top of it. Note that the shapes of the 4.2 K MDs do not depend on the $V_{geq}$ value: if the same protocol is applied under a different $V_{geq}$, the same shapes are observed after the cooling, the MDs being then centered on the new $V_{geq}$ value.

Building on this uniformity in shape of the MDs, we can characterize the influence of the application of $V_{geq}$ on a given sample by the width of the MD. For the two most resistive samples A1 and B1, 4.2 K MDs are fully visible in the 60 V window of the experiment and their widths at half maximum $\Delta V_g$ can be deduced directly from the top panels of Figure 6. For the other samples, the 4.2 K MDs extend to larger $V_g$ values (see the remaining slope $d\Delta G/dV_g$ at $-30$V and 30V) and a more subtle procedure needs to be applied. As it is shown in the lower panels of Figure 6, the different 4.2 K MDs corresponding to the various Nb content for a given thickness

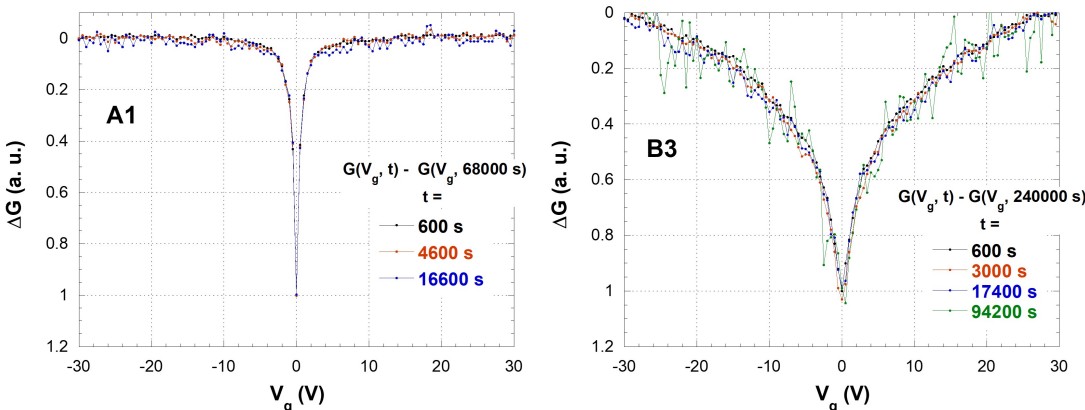

Figure 5: Time independence of the MD shape after a cooling from room temperature to 4.2 K ($V_{geq} = 0$ V). The plotted curves correspond to the difference between the $G(V_g)$ curves measured a time $t$ after the cooling, and the last $G(V_g)$ curve measured (long-time reference). These difference curves are shifted to 0 at large $V_g$ values in order to remove the background drift discussed in Section 4 and the value at 0 V ($V_{geq}$) is normalized to 1 in order to compare the MD shapes (the vertical axes are reversed).

Table 2: Widths at half maximum $\Delta V_g$ and screening length values $L_{sc}$ for the different a-NbSi films.

| Name | Thickness | % Nb | $R_{s4K}$ | $\Delta V_g$ | $L_{sc}$ |
|------|-----------|------|-----------|--------------|----------|
| A1 | 2.5 nm | 13 | 97 MΩ | 0.7 V | $\geq 2.5$ nm |
| A2 | 2.5 nm | 14 | 7.0 MΩ | 2 V | $\geq 2.5$ nm |
| A3 | 2.5 nm | 16 | 330 kΩ | 12 V | $\geq 2.5$ nm |
| B1 | 12.5 nm | 7 | 53 MΩ | 2.5 V | $\simeq 7$ nm |
| B2 | 12.5 nm | 8 | 970 kΩ | 7 V | $\simeq 4$ nm |
| B3 | 12.5 nm | 10 | 28 kΩ | 12 V | $1.3 - 1.9$ nm |

can be overlaid by using ad hoc vertical and horizontal ($V_g$) scaling factors. $\Delta V_g$ estimates for A2 and A3 (resp. B2 and B3) can thus be obtained by multiplying A1- (resp. B1-) $\Delta V_g$ by the corresponding $V_g$ scaling factors. The $\Delta V_g$ values of our NbSi films are displayed in Table 2 and found to vary between 0.7 V and 12 V. If we scale the values reported in the literature for other systems so that they correspond to a 100 nm-thick $SiO_2$ gate insulator[1], we get: 0.2 V – 2 V in InOx films [12,13], 1 V – 2 V in 20 nm thick granular Al films [28], 0.1 V in Be films [15], 1 V in $Tl_2O_{3-x}$ films [16] and 0.6 V - 2 V in Ge-Te alloys [17–19]. The value of 12 V found in films A3 and B3 is by comparison extremely large, while the MD width in film A1 is similar to values observed in 10 nm thick granular Al films having similar $R_s$. We should note that the existence of a frozen contribution to the MD is usually overlooked in the literature, although it might influence the determination of the MD width if care is not taken to only measure the contribution of the narrow growing MD. For example in Be films, only a narrow central part of the MD close to $V_{geq}$ is modified when the sweeping parameters are changed (see Figure 5 of Ref. [15]), which indicates that the measured total MD may results from the superposition of a wide frozen contribution and a narrower time-dependent one.

How can one interpret the MD width variations we observed? As can be seen in Figure 7, in

---

[1]For a gate insulator with a dielectric constant $\epsilon$ and a thickness $d$, $\Delta V_g$ is multiplied by $(\epsilon/4) \times (100 \ nm/d)$.

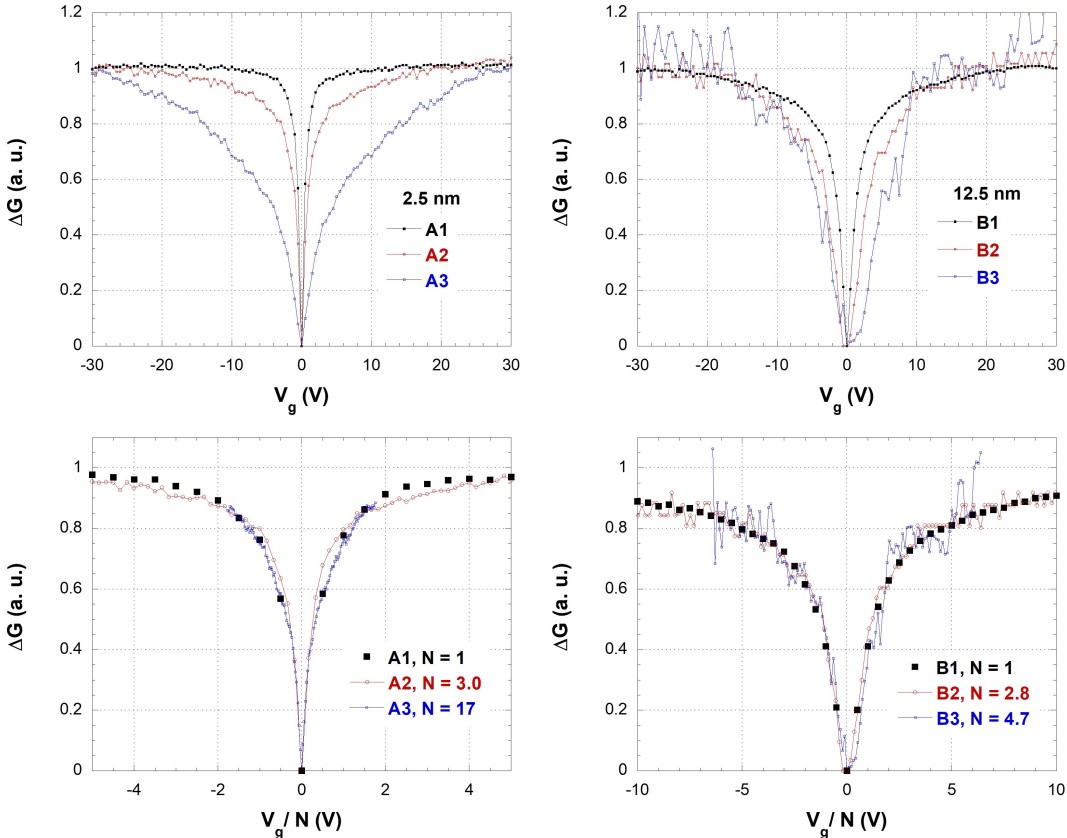

Figure 6: Shape of the 4.2 K contributions $\Delta G$ to the MD as a function of $V_g$ (top panels) and of $V_g/N$ (bottom panels). Top panels: $\Delta G$ was fixed to 1 at 30 V and 0 at 0 V for all the samples. Bottom panels: vertical and horizontal scalings were applied for A2 and A3 (resp. B2 and B3) curves so that they can be superimposed on that of A1 (resp. B1). The numbers indicated for $N$ are the factors by which the A2 and A3 (resp. B2 and B3) $V_g$ scales were compressed. See the text for details. Left side panels: 2.5 nm thick films. Right side panels: 12.5 nm thick films. $V_{geq} = 0$ V for all the films.

12.5 nm-thick films for which the screening length values $L_{sc}$ can be estimated (see Section 4), a correlation $\Delta V_g \propto 1/L_{sc}$ is found between $L_{sc}$ and the MD width. Interestingly enough, the extrapolation of NbSi data to thinner MDs using this law includes granular Al results [28] and predicts a screening length larger than $\simeq 60$ nm for the InOx films of Ref. [29], in rough agreement with the absence of background relaxations in films thinner than 75 nm. If we assume that the screening is provided by the electrons populating an Efros-Shklovskii Coulomb gap [44,45], the density of thermally excited electrons at a finite $T$, $n(T)$, is given by:

$$n(T) \simeq \alpha/L_{sc}^3(T), \tag{1}$$

where $\alpha$ is a constant of order unity. In the percolation approach of the MD proposed by Lebanon and Müller [20], the instability scale $\Delta V_g$ for the conductance is obtained when the density of carriers induced by the gate voltage change, $n(\Delta V_g)$, becomes larger than $n(T)$. For films thicker than $L_{sc}$:

$$n(\Delta V_g) \simeq (C\Delta V_g)/L_{sc}, \tag{2}$$

where $C$ is the capacitance per area between the metallic gate and the film. According to

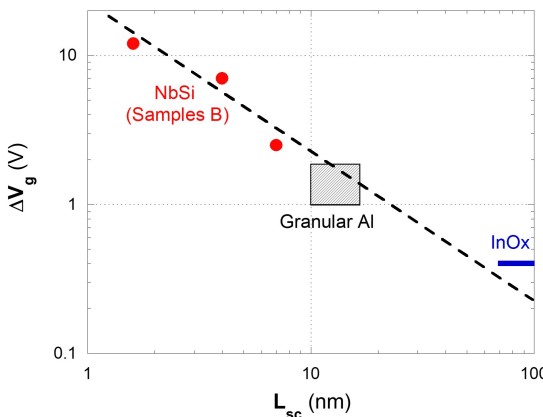

Figure 7: $\Delta V_g$ versus $L_{sc}$ for a-NbSi (samples B), granular Al (Ref. [28]) and InOx films (samples of Figure 8 Ref. [29]). The doted line represents the law $\Delta V_g \propto 1/L_{sc}$.

Equations 1 and 2, the instability criterion $n(\Delta V_g) = n(T)$ thus leads to:

$$\Delta V_g = [n(T)L_{sc}]/C \simeq (\alpha/C)(1/L_{sc}^2). \tag{3}$$

This prediction disagrees with the experimental data of Figure 7, which suggests that either the samples are not in the Efros-Shklovskii Coulomb gap regime (the sample B3 is indeed very close to the metal-insulator transition), or the instability criterion for the MD should be reconsidered. Note that Equation 3 predicts a $T^2$ dependence for $\Delta V_g$ ($L_{sc} \propto 1/T$), instead of the $T$ dependence experimentally observed [2].

In InOx films, the MD width was shown to depend systematically on $n$, the charge carrier density deduced from Hall effect measurement at room $T$, but not on the $R_s$ value of the films [13]. In our system, we find a seemingly contradictory result, namely the fact that the MD width is determined by the sample's resistance. More precisely the quantity $\Delta V_g/\min(L_{sc}, T_h)$ correlates very well with $R_s(4.2 \text{ K})$, as shown in Figure 8. This quantity represents the charge density one must induce in the unscreened part of the sample in order to push it out of equilibrium. But some other elements may moderate this discrepancy between NbSi and InOx films. First, it was recently suggested [42] that in InOx the $n$ dependence may not be a proof of a dependence on the carrier density per se, but rather on the disorder strength. Indeed both are believed to be closely related in samples that are close enough to the metal-insulator transition (MIT), which is presumably the case for insulating samples having measurable resistances at low temperature. Second, in NbSi films, the interrelationship between $R_s$, the chemical composition, the thickness and the charge carrier density $n$ is still poorly understood. However, it seems reasonable to assume that for a given thickness, the smaller $R_s$ (the larger the Nb content), the larger $n$. In 100 nm-thick films, a direct estimate of $n$ by Hall effect measurements gives $n = 3.9 \times 10^{23} e/\text{cm}^3$ for a metallic film with 26% of Nb [46], compared to a value $5 \times 10^{22} e/\text{cm}^3$ obtained from the linear term of specific heat for a weakly insulating film with 9% of Nb [34]. Note that this last value is larger than the highest $n$ reported in InOx films, in qualitative agreement with the large MD widths observed in low $R_s$ NbSi films.

# 6 Memory dip dynamics

In spite of the large differences in their widths, the 4.2 K MDs all grow in a similar (log) way with the time elapsed since the cooling in all samples studied, with no signs of saturation up to

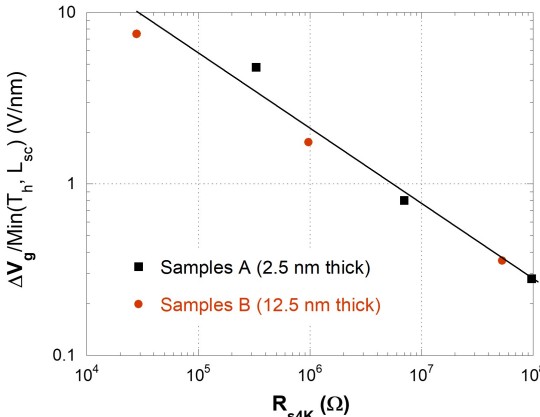

Figure 8: $\Delta V_g/T_h$ for samples A, $\Delta V_g/L_{sc}$ for samples B, as a function of $R_s$ at 4.2 K. See the text for details.

the longest times achieved ($\simeq 2 \times 10^5$ s, see Figure 9). Such a logarithmic time dependence is commonly observed in all other disordered insulating systems where electrical glassy features have been found [1]. It can be simply described as resulting from the sum of independent degrees of freedom having a log-normal distribution of relaxation times [2,47,48]. Note that this logarithmic relaxation includes sample B3 which has a rather small $R_s$ value of $28k\Omega$ at 4.2 K, putting it very close to the MIT: the amplitude of the MD is very small but the time scales over which the relaxation is measurable are large. This shows that, as we already stressed in another context [27], the amplitudes of the conductance relaxations cannot directly quantify the dynamics and allow comparisons between systems (like for instance in [29]). Actually one expects that approaching the MIT from the insulating side, the sensitivity of our experimental probe (the conductance) to any changes occurring in the sample, does vanish. Only direct measurements of relaxation times are meaningful. These were only recently achieved in InOx films [12] and phenomenologically showed that both low $R_s$ and low carrier densities ($n \leq 10^{19}e/\text{cm}^3$) are needed to get measurable shorter relaxation times. The carrier density value of $5 \times 10^{22}e/\text{cm}^3$ deduced from the linear term of the specific heat in a weakly insulating NbSi film [34] suggests that our B3 sample, although of rather small resistance, has a charge carrier density that is most probably too large for the higher time cutoff of the logarithmic relaxation to occur within measurable times.

The dynamic response to a $V_g$ change was tested using the so-called erasure protocol. The sample is first prepared in a given initial state by cooling it from room $T$ to 4.2 K under $V_g = V_{g0}$ and by maintaining it under this $V_g$ value for a time $t_a$ (step 0). Then, $V_g$ is changed to $V_{g1}$ and a new MD centred on $V_{g1}$ is formed during a time $t_w$ (step 1). Last, $V_g$ is changed to $V_{g2}$ and the erasure of the $V_{g1}$ MD is measured as a function of time (step 2). In Figure 10 (and 11), $V_{g0} = V_{g2} = 0$ V and $V_{g1} = 20$ V. At constant time intervals, $G$ is measured at $V_g = -20$ V (reference value), 0 V ($V_{g2}$ value) and 20 V ($V_{g1}$ value) and $[G(-20\text{V}) - G(20\text{V})]/G(-20\text{V})$ is taken as the relative MD amplitude $\Delta G/G$ of the 20 V MD [2]. In granular Al films, it was shown that if all the protocol was performed at a fixed $T$ (isothermal protocol) and if $t_a \gg t_w$, the erasure curve of the MD formed at $V_{g1}$ is well described by the law:

$$\Delta G(t, t_w) = A(T)\ln(1 + t_w/t). \tag{4}$$

---

[2]Note that defined in this way, $\Delta G/G$ doesn't go to zero when the time goes to infinity since there is always a time independent difference between the conductance at -20 V and 20 V, coming for example from a normal field effect contribution. This asymmetry was measured during the initial cooling under $V_{geq} = 0$ V (no MD at $V_{g1}$) and subtracted from the $\Delta G/G$ values.

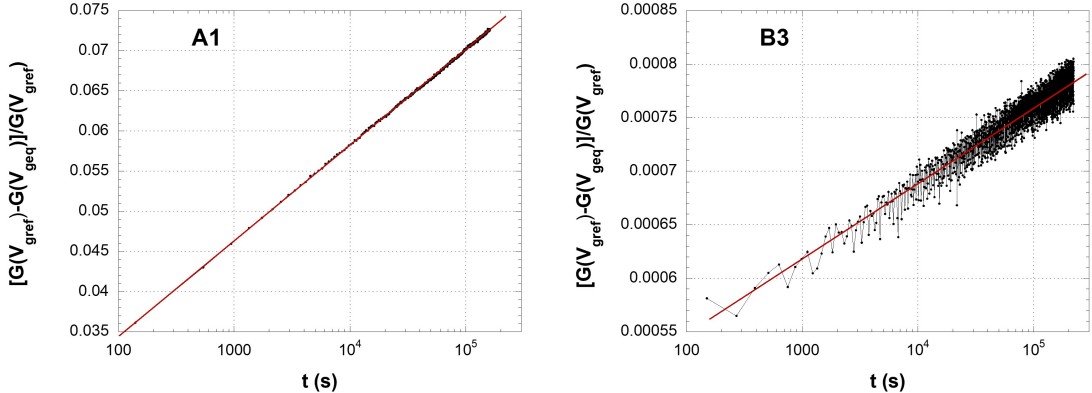

Figure 9: Growth of the MD amplitude as a function of the time elapsed after a cooling to 4.2 K. The conductance is measured at constant time intervals under $V_{gref} = -20$ V, and compared to the $V_{geq} = 0$ V value. Left side panel: sample A1; right side panel: sample B3. In sample B3, the large noise level is due to the smallness of the MD (the MD amplitude is smaller than 0.1% after $2 \times 10^5$ s).

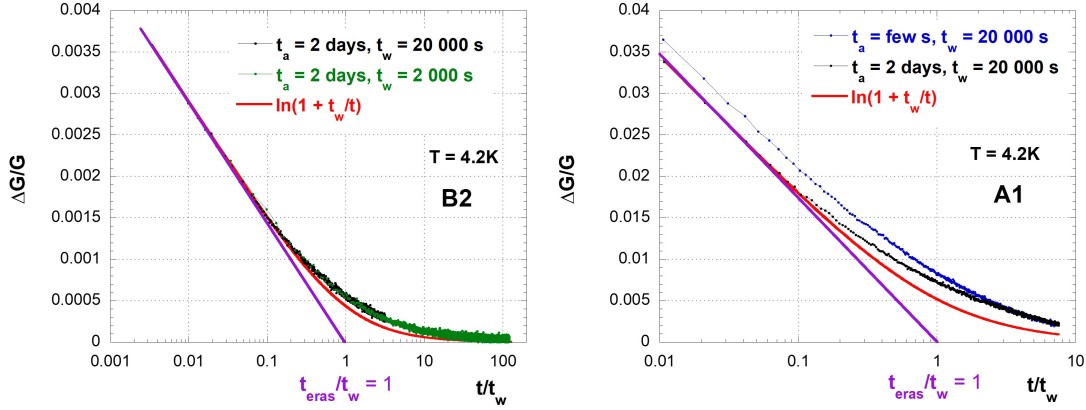

Figure 10: Influence of the waiting times $t_a$ and $t_w$ on the 4.2 K erasure curves for samples B2 (left panel) and A1 (right panel). See the text for details.

Equation 4 was shown to result from a log-normal distribution of relaxations times switching back and forth under $V_g$ changes [2, 48]. The curve thus scales with $t_w$ and the typical erasure time $t_{eras}$, given by the intercept of the logarithmic dependence at short times with the $\Delta G = 0$ line, is equal to $t_w$. Note that in granular Al films, departures from Equation 4 are observed when $t_a < t_w$ and when $t_{eras}$ becomes larger than $t_w$ [5]. This dependence of the dynamics with the time spent since the cooling of the sample, i.e. the age of the system, a property called ageing, is the hallmark of glasses. In our NbSi films, we observed significant and unexplained departures from the analytical time dependence of Equation 4 around $t \simeq t_w$ (see Figure 10) but the $t_w$ scaling and ageing effects are still present.

In its isothermal form, the erasure protocol described above cannot reveal the $T$ dependence of the dynamics [2]. In order to do so, one has to use a different version implying two temperatures [2, 25, 32]: the MD formation (step 1) is performed at $T_1$, and the MD erasure (step 2) at $T_2 \neq T_1$ (the $V_g$ change from $V_{g1}$ to $V_{g2}$ is done just before the $T$ change from $T_1$ to $T_2$). One wants to know whether the time needed to erase the $V_{g1}$ MD depends on the respective values of $T_1$ and $T_2$. The results of such a protocol for $T_2 = 4.2$ K and $T_1 = 4.2$ K,

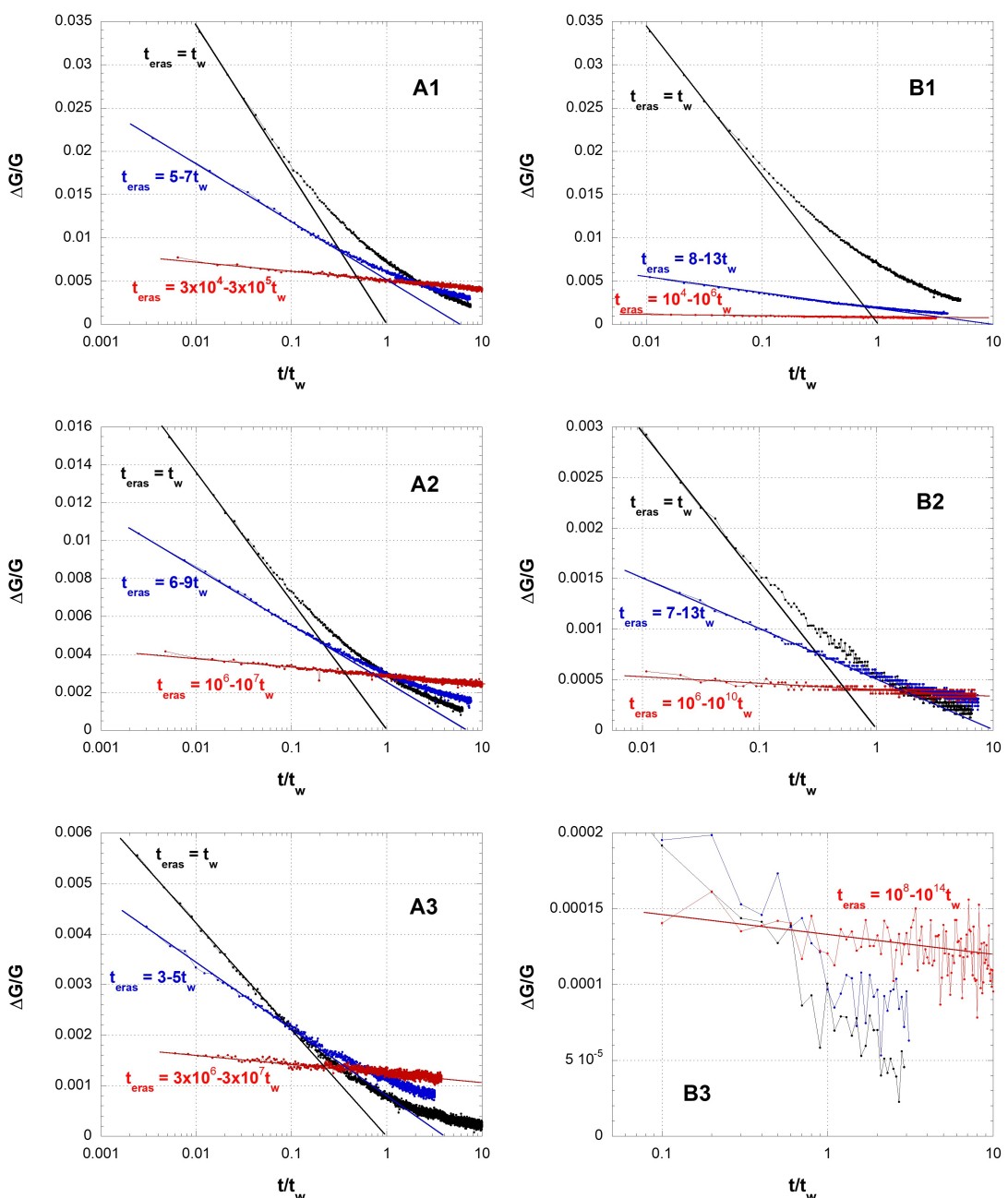

Figure 11: 4.2 K erasures of MDs formed during a time $t_w = 20000$ s at different temperatures $T_1$. In our set-up the cooling takes $\simeq 30$ s from 9 K and $\simeq 100$ s from 20 K. Intersections between the straight lines and the $\Delta G/G = 0$ line give an estimate of the typical erasure times $t_{eras}$. Note that due to the smallness of the MD amplitudes, the noise level is high for the sample B3 data. The relaxation measurements also start at longer times so that the straight lines of the $T_1 = 4.2$ K and 9 K erasure curves cannot be reliably drawn.

9 K and 20 K are plotted in Figure 11. The samples were let for at least one day at 4.2 K before starting the protocol in order to minimize the ageing effects described above.

For each erasure curve of Figure 11, we can define $t_{eras}$ as the intercept between the time axis and the short time linear parts of the curves ($t/t_w < 0.03$ for $T_1 = 4.2$ K, $t/t_w < 0.3$ for $T_1 = 9$ K and almost all the time window for $T_1 = 20$ K). The dependence of $t_{eras}$ with $T_1$

Table 3: Activation energies $\Delta E$ deduced from the erasure times of Figure 11 ($T_2 = 4.2$ K). See the text for details.

| Name | Arrhenius law hypothesis | |
| --- | --- | --- |
| | $\Delta E$ ($T_1 = 9$ K) | $\Delta E$ ($T_1 = 20$ K) |
| A1 | 13 K - 15 K | 50 K - 70 K |
| A2 | 14 K - 17 K | 70 K - 90 K |
| A3 | 9 K - 13 K | 80 K - 90 K |
| B1 | 16 K - 20 K | 50 K - 70 K |
| B2 | 15 K - 20 K | 70 K - 120 K |
| B3 | | 100 K - 170 K |

reflects the temperature dependence of the conductance relaxation dynamics. If $T_1 = 4.2$ K, we get the usual isothermal result $t_{eras} = t_w$ while a marked increase of $t_{eras}$ is observed when $T_1 > 4.2$ K: depending on the samples, $t_{eras}$ varies between $4t_w$ and $10t_w$ for $T_1 = 9$ K and between $\simeq 10^5 t_w$ and $\simeq 10^{11} t_w$ for $T_1 = 20$ K. This clearly shows that the dynamics is thermally activated and slows down upon cooling.

The comparison of the different MD amplitudes provide a natural explanation for the various aspects of the curves shown in Figure 2. Looking at the curves of Figure 11 at short times, one sees that depending on the samples, the amplitudes of the MDs formed at $T_1 > 4.2$ K compare very differently with the ones formed at 4.2 K : for a given thickness, the smaller the $R_s$, the larger they are compared to 4.2 K MDs. This trend is especially pronounced in 12.5 nm thick films (B films). Consequently, when the high temperature MDs are much smaller than the 4.2 K contributions (film B1), the "frozen" MD accumulated during the cooling under $V_{geq} = 0$ V is almost invisible, thus the flat background. In opposite cases, the "frozen" MD can be the dominant contribution and the growing 4.2 K contribution can be hardly discernable (see film B3 in Figure 2).

We have tried to explain quantitatively the erasure curves of Figure 11, assuming that the same modes are relaxing back and forth during the $V_g$ changes at $T_1$ and $T_2$. If the relaxation times obey an Arrhenius dependence of the form $\tau_i(T) \propto \exp(\Delta E/T)$ with a single activation energy $\Delta E$, then $t_{eras}/t_w = \exp[\Delta E(1/T_2 - 1/T_1)]$. But as it is seen in Table 3, the $\Delta E$ values corresponding to $T_1 = 20$ K data are from 4 to 10 times larger than the ones corresponding to $T_1 = 9$ K. Such a simple Arrhenius model can thus not describe the dramatic increase of the erasure times observed in a given sample when $T_1$ is changed from 9 K to 20 K. We have tested other simulations with a distribution of activation energies, but without being able to replicate the experimental data of Figure 11. The $T$ dependence dynamics of the NbSi films is thus non trivial and would require a deeper change in our theoretical assumptions.

The non isothermal erasure protocol was also recently applied to granular Al films below 30 K and reveal a qualitatively similar slow down of the dynamics when $T$ is lowered [25]. However, the $T$ variation of the slow dynamics was found to be essentially of the Arrhenius type, with an activation energy of the order of 30 K which does not depend much on the resistance of the films. In discontinuous gold films, a different protocol has been used: the erasure of a MD formed at $T_1$ and during the cooling to $T_2$ is measured at $T_2$ and compared with the formation of a new isothermal MD at $T_2$ [22, 23]. Signs of a strong temperature dependence of the dynamics are present since the MDs formed upon cooling between $T_1$ and $T_2$ cannot be significantly erased by subsequent $V_g$ changes at low temperatures, even after days of measurements. But the differences in the protocols make the comparison with amorphous NbSi and granular Al films difficult. In InOx, the situation is still unclear. No temperature dependence or even an acceleration of the dynamics when the temperature is lowered were reported [13, 26] but we have questioned the protocols used during these experiments [25].

Our results clearly stress that the activated character of the dynamics is not limited to granular or discontinuous systems and call for reconsidering InOx results. In order to clarify the experimental situation and to make progress on theoretical models, the erasure protocols described in the present study should be applied to the other systems in which electrical glassy effects have been found and in a $T$ range as large as possible. Preliminary results show that the MD dynamics is indeed also thermally activated in amorphous InOx films [49] .

## 7 Conclusion

In conclusion, our field effect measurements on amorphous NbSi films lead to the following results. First, we observed significant background conductance relaxations in 12.5 nm thick films, which reflect a spatial extent of the gate voltage disturbance of only few nanometres. We interpret this length as the screening length of the films and we suggest that the difference with the lower bound of 75 nm reported in amorphous indium oxide films stems from the chemical inhomogeneity of the films giving rise to electrical inhomogeneities. Second, the widths of the memory dips formed at 4.2 K vary strongly with the film parameters. They correlate with the $R_s$ values of the films, the smaller $R_s$, the wider the dip, and with the screening length of the films, which may help to better understand the physical mechanisms involved. Third, the conductance relaxations after cooling are logarithmic in time over days with no cutoff of the time distribution visible, even in low $R_s$ films in close vicinity to the metal-insulator transition. When triggered by $V_g$ changes, the relaxations display ageing and scale with the waiting time spent under a new $V_g$ value. Last, the temperature dependence of the glassy dynamics was tested between 4.2 K and 20 K by applying a non-isothermal erasure protocol. We found in all the films a strong slowdown of the dynamics upon cooling, in qualitative agreement with recent results in granular Al films. The thermally activated character of the dynamics is thus not limited to granular or discontinuous films, which calls for a reexamination of the $T$ dependence of the dynamics in indium oxide films. More generally, our results demonstrate that there is no clear line between continuous and discontinuous systems, which is a major step towards an universal vision of the electron glass effects.

## Acknowledgements

We gratefully acknowledge discussions with M. Müller.

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
