# Peer review of "Electron glass effects in amorphous NbSi films"

_SciPost Physics, doi:SciPost Phys. 8, 056 (2020)_

## Round 1 · Referee Report · Anonymous · 2020-2-25

Strengths

1. Addresses several key experimental questions in our understanding of electron glasses.

2. High-quality experimental data.

3. Thorough and complete analysis of the dependence of the shape of the memory dip on time and on experimental parameters.

4. Excellent analysis of the experimental results using theoretical arguments, and comparison with experiments on indium oxide.

Weaknesses

1. The graphical quality of the plots could be improved.

Report

This very interesting paper presents a thorough experimental study of the "memory dip" (MD) in continuous insulating films of amorphous NbSi, of varying thickness and resistance. The MD has been studied before in granular/discontinuous films and in indium oxide. Here the authors show that many qualitative features of the MD previously often assumed to only apply to granular/discontinuous films are also present in this continuous amorphous material. I think this is a significant step forward in the understanding of electron glasses and will no doubt stimulate further research.

The paper is clearly written, and the interpretation of the experimental results is very careful. I only have a few minor comments / questions.

1. The analysis leading to Eq.3 assumes that the screening length depends on the temperature as L_{sc} ~ 1/T. It seems to me that an experimental verification of this assumption would be quite interesting. Have the authors considered repeating the analysis contained in Fig.3 for different temperatures, in order to extract this dependence? (I do realize that the range of L_{sc} is quite limited, though.) Perhaps this may point to a solution of the discrepancy between the theoretical prediction and the observed dependence of \Delta V_g on L_{sc} and T.

2. It would be interesting to give the approximate values of T_0 in Fig.1. Assuming an Efros-Shklovskii scenario, one could then estimate the number of carriers within a localization radius, and compare it with that of other
disordered insulators, in particular indium oxide.

3. In the Introduction, the authors state that "the length over which the glassy features are perturbed upon the application of a gate voltage change" will be referred as "penetration length". However, in the rest of the paper the term "screening length" is used instead.

4. Again in the Introduction, the authors mention recent numerical findings of an equilibrium transition. It may be worth adding, for the benefit of the reader, that the transition temperature found in Ref.11 is exceedingly low compared with the Coulomb interaction (much lower than the already low
transition temperature predicted by mean field theory).

5. Can the authors give a rough estimate of the experimental error in the curves G(V)? Have they tried more than one sample for a given thickness and composition?

6. The symbol R_s should be defined the first time it is used (in the Introduction).

7. It seems to me that in Eq.2, C should be the capacitance per unit area.

8. A few typos: which -> whose (p.1); extra "m" in mm symbol (p.3); thereafter -> hereafter (p.4); initial cool -> initial cooling (p.8); slow down -> slows down (p.16); MDSs -> MDs (p.16).

  • validity: high
  • significance: high
  • originality: good
  • clarity: good
  • formatting: reasonable
  • grammar: good

Author:  Julien Delahaye  on 2020-03-18  [id 765]

(in reply to Report 1 on 2020-02-25)

  1. As the referee rightly points, an experimental verification of the relation between the temperature and the screening length around 4K would be of great value. But unfortunately, the samples presented in this study are not favourable to this proposal. Indeed, the dip width of the most resistive sample 12.5nm thick (sample B1) is already so large at 4.2K that it reaches the strictly Vg independent regime around 30V. Since the dip width is even larger at higher T (it grows roughly as T), an accurate determination of the screening length in a large enough T range will be very imprecise. In less resistive films, in addition to this problem, the screening lengths are already very short at 4.2K (few nanometres) and a limited range of values is thus available from higher T measurements. It seems to us that the only way one can determine the relation between Lsc and T would be to measure more restive films (the dip should be thinner, and the screening length larger), or to extend the relaxation measurements to temperatures smaller than 4K. The suggestion of the referee was however applied between 4K and 10K to the more favourable case of granular Al films (see Ref. 38). But even there, if the relation Lsc ∼ 1/T is in agreement with the experimental data, the uncertainties on the Lsc values are such that another close T dependence cannot be excluded (like Lsc ∼1/T^(1/2)).
  2. We chose deliberately not to give any quantitative parameters associated with the resistance versus temperature curves of our NbSi films. And if the law R ~ exp(T0/T)^0.5 gives a good description of the R - T divergence between 40K and 4K for samples A3 and B2 (but not for the other ones !), its physical interpretation is problematic. The T0 values extracted from this law are respectively 57K (A3) and 86K (B2). According to the usual ES interpretation, the localisation length xi is given by beta e^2/(epsilon kBT0) where beta is a numerical constant (2.8 in 3D) and epsilon is the dielectric constant of the material. If we take epsilon = 11.7epsilon0 (bulk Si value), we get very large localisation lengths of 900nm (A3) and 600nm (B2). But these values should be taken with great care. First, the resistance divergence observed for these two films remains limited (about one order of magnitude only), which is quite small in order to establish the exponent α of a law R ~ exp(T0/T)^α (a temperature dependent prefactor can give a significant contribution). Second, the dielectric constant of the material should diverge at the metal-insulator transition, and it is thus probably much larger than that of bulk Si in films A3 and B2. Last, some studies suggest that due to the contribution of many electron effects, the beta value of 2.8 is largely underestimated (see for example Massey et al, Phys. Rev. B 62, R13270 (2000)). For all these reasons, we prefer not to propose an interpretation of the R-T laws observed in Fig. 1, but simply use them as an indication of their insulating character.
  3. The term “penetration length” is neutral, while the term “screening length” relies on a physical interpretation. Thanks to the referee remark, we have added a sentence in part 4 where this problem is discussed in details in order to clarify this point: “Note that according to this interpretation, the penetration length introduced earlier is nothing else than the screening length of the system.” This difference is also stressed in the conclusion: “We interpret this (penetration) length as the screening length of the films…”.
  4. We have supplemented the sentence: “The occurrence of such a transition was recently found in numerical simulations of the 3D Coulomb glass model” with “but with a transition temperature much lower than the mean field prediction”. Ref. 11 was also updated (Condmat preprint to Phys. Rev. B).
  5. We have measured only one sample for given thickness and composition (the uncertainty is about 0.1% for the absolute composition and 0.5nm for the thickness). Concerning the experimental error in the G(Vg) curves, it is not really the noise level on the conductance which is important, but how this noise level compares with the amplitudes of the quantities of interest (conductance relaxations, memory dip, etc.). An estimate of the conductance “error” can be inferred from scattering of the conductance points around the straight lines in Fig. 9. For film A1, the experimental error on G is of about 0.01%, whereas the conductance relaxation amplitude over three orders of magnitude in time is 2.5%, 250 times larger. Consequently, the noise is almost invisible in this film. For film B3, the experimental error on G is smaller in relative value, about 0.002%, but the conductance relaxation over three order of magnitude in time lies now below 0.02%: large fluctuations of the conductance are thus visible when the relaxation of this film is measured (see also the same film on Fig. 5, 6 and 11).
  6. It has been done.
  7. It has been corrected.
  8. They have been corrected.

---

## Round 1 · Referee Report · Anonymous · 2020-3-5

Strengths

1 - Thorough study of the glassy properties in NbSi, to allow additional insights into the electron glass in general, and in relation to their properties in granular systems and other amorphous systems.

2 - Various issues, including background relaxation and its relation the screening length, and temperature dependence of the memory dip dynamics and its relation to activation are addressed.

Weaknesses

1 - The electron glass is a very complex system, for which experimental data has been given numerous competing interpretations. The authors interpretations of their own results may be put in question, and their statements on the InOx, while stated very strongly, are certainly debatable.

Report

In this paper the authors study the NbSi electron glass, as function of film thickness and resistance. It is found that the memory dip widens with the decrease of resistance, and that background relaxation is pronounced in the thick sample, a fact attributed to the screening length being smaller than the sample width. The authors then relate the structure of the memory gap to the screening length of the system, using the model of Lebanon and Muller. Still, the authors make strong statements about the screening length not only in their studied system, but also in InOx. I would expect the authors to be more modest in their interpretation of their results, certainly in their interpretation of other systems.
The authors then study the dynamics of the memory dip. Their experimental findings of temperature dependence more pronounced than expected from simple activation are intriguing, calling for further theoretical and experimental effort.

Requested changes

My suggestion to the authors is to phrase their interpretation in a way that makes it clear that it is debatable, especially when discussing systems not measured in this work.

  • validity: good
  • significance: ok
  • originality: good
  • clarity: good
  • formatting: good
  • grammar: good

Author:  Julien Delahaye  on 2020-03-18  [id 766]

(in reply to Report 2 on 2020-03-05)

We agree with the referee that most of the issues discussed in this article have been and are still a matter of debate (penetration length, temperature dependence of the dynamics, …), and this is why we tried to be cautious in our discussions (which was noticed by the other referee: “the interpretation of the experimental results is very careful”). For example, concerning the penetration length interpretation, we used the terms: “Other experimental issues remain controversial or are waiting for a satisfactory explanation” (Introduction) ; “Another debated issue….” (Introduction); “…which in our picture would imply” (Section 4) ; “We interpret this length as the screening length…and we suggest…” (Conclusion). In order to follow the referee recommendations, we have done the following changes: - the sentence (Section 4): “All these findings thus suggests that InOx films have a large screening length compared to granular Al and NbSi films, which comes from a material specific long range electrical inhomogeneity” is replaced by a less affirmative one: “All these findings suggest that a material specific long range electrical inhomogeneity may explain why the penetration length is larger in InOx than in granular Al and NbSi films.” - we add at the end of the penetration length discussion in Section 4: “These controversial results emphasize that the physical origin of the penetration length, and especially of the large values observed in InOx films, remains an open and debated issue. Screening length determination by other means than gate voltage induced conductance relaxations would be very helpful in order to confirm our interpretation.” Concerning the dynamics, we also agree with the referee on the fact that our temperature dependence is intriguing and deserves further experimental and theoretical efforts. This is the meaning of the sentence before Table 3 (Section 6): “The T dependence dynamics of the NbSi films is thus non trivial, and would require a deeper change in our theoretical assumptions”.

---

## Round 2 · Author Response

Here is our latest version of the article which takes into account the referees remarks. See also our response to referee reports.

---

## Round 2 · List of Changes

• Section 1 (Introduction): last sentence of paragraph 3 (added): "...but with a transition temperature much smaller than the mean field prediction."
  • Section 4: middle of paragraph 2 (added): "Note that according to this interpretation the penetration length introduced earlier is nothing else than the screening length of the system."
  • Section 4: end of paragraph 3 (replacement): "All these findings suggest that a material specific long range electrical inhomogeneity may explain why the penetration length is larger in InOx than in granular Al and NbSi films."
  • Section 4: end of paragraph 4 (added): "These controversial results emphasize that the physical origin of the penetration length, and especially of the large values observed in InOx films, remains an open and debated issue. Screening length determination by other means than gate voltage induced conductance relaxations would be very helpful in order to confirm our interpretation." We have also improve the resolution of the Figures.

---

## Editorial Decision

published